

# Arboreal twig-nesting ants form dominance hierarchies over nesting resources

Senay Yitbarek[1] and Stacy M. Philpott[2]

[1] University of California, Berkeley, Berkeley, CA, United States of America
[2] University of California, Santa Cruz, Santa Cruz, CA, United States of America

## ABSTRACT

Interspecific dominance hierarchies have been widely reported across animal systems. High-ranking species are expected to monopolize more resources than low-ranking species via resource monopolization. In some ant species, dominance hierarchies have been used to explain species coexistence and community structure. However, it remains unclear whether or in what contexts dominance hierarchies occur in tropical ant communities. This study seeks to examine whether arboreal twig-nesting ants competing for nesting resources in a Mexican coffee agricultural ecosystem are arranged in a linear dominance hierarchy. We described the dominance relationships among 10 species of ants and measured the uncertainty and steepness of the inferred dominance hierarchy. We also assessed the orderliness of the hierarchy by considering species interactions at the network level. Based on the randomized Elo-rating method, we found that the twig-nesting ant species *Myrmelachista mexicana* ranked highest in the ranking, while *Pseudomyrmex ejectus* was ranked as the lowest in the hierarchy. Our results show that the hierarchy was intermediate in its steepness, suggesting that the probability of higher ranked species winning contests against lower ranked species was fairly high. Motif analysis and significant excess of triads further revealed that the species networks were largely transitive. This study highlights that some tropical arboreal ant communities organize into dominance hierarchies.

## INTRODUCTION

A long-standing goal in ecology has been to determine the underlying mechanisms that give rise to species coexistence in local communities, especially in assemblages with multiple competing species (*MacArthur, 1958*; *Hutchinson, 1959*). Numerous mechanisms have been proposed for maintaining species coexistence (*Wright, 2002*; *Silvertown, 2004*). Interspecific competitive trade-offs, whereby the dominance of a particular species in one environment is offset by the dominance of another species in a different environment, can lead to spatial segregation between species (*Tilman, 1994*; *Levine, Adler & Yelenik, 2004*). These interspecific interactions are thought to lead to the long-term stable coexistence of ecologically similar species (*Levins, 1979*; *Holt, Grover & Tilman, 1994*; *Chesson, 2000*;

Corresponding author
Senay Yitbarek, senay@berkeley.edu

*Bever, 2003*; *Rudolf & Antonovics, 2005*), and may also be characterized by dominance hierarchies. Dominance hierarchies have been observed in a wide range of taxa, from vertebrates to invertebrates (*Chase & Seitz, 2011*). Species can be ranked into a hierarchy based on their behavioral dominance during interspecific competitive encounters for resources (*Davidson, 1998*). For example, dominance ranking was positively associated with body mass in bird species, with heavier species more likely to monopolize food sources in contrast to lighter species (*Francis et al., 2018*). However, dominance rankings can be determined by many other factors including age, sex, aggressiveness, and previous encounters (*Haley, Deutsch & Le Boeuf, 1994*; *Zucker & Murray, 1996*). Furthermore, interspecific dominance hierarchies have been used to understand patterns of local species coexistence in ecological communities (*Morse, 1974*; *Schoener, 1983*).

In ant communities, dominance hierarchies have been used to examine interspecific tradeoffs that may explain species coexistence patterns (*Stuble et al., 2013*). These trade-offs include the discovery-dominance trade-off, the discovery-thermal tolerance tradeoff, and the discovery-colonization trade-offs (*Cerdá, Retana & Manzaneda, 1998*; *Stanton, Palmer & Young, 2002*; *Lebrun & Feener, 2007*; *Stuble et al., 2013*). In addition to testing interspecific trade-offs, dominance hierarchies have been used to understand the role of dominant species in structuring local communities and species composition, such as partitioning dominant and subdominant species within guilds (*Baccaro, Ketelhut & Morais, 2010*; *Arnan, Cerdá & Retana, 2012*). Dominant ant species can play an important role in the structuring of local communities. For example, *Formica* species dominating a boreal ecosystem divert resources away from subdominant competitors (*Savolainen & Vepsäläinen, 1988*). In Mediterranean ecosystems, subdominant species forage at nearly lethal environmental conditions while dominant species reduce their own mortality risk by foraging at more favorable temperatures (*Cerdá, Retana & Manzaneda, 1998*; *Castillo-Guevara et al., 2019*). In tropical ecosystems, competing arboreal ants can be structured into a dominance hierarchy with higher ranked ant species having greater access to nesting sites and extrafloral nectaries (*Blüthgen, Stork & Fiedler, 2004*; *Díaz-Castelazo et al., 2004*). However, levels of uncertainty associated with outcomes of interspecific interactions between ants are often not quantified (*Stuble et al., 2017*). Furthermore, its remains unclear how arboreal ants or tropical ants are structured at the community level, such as when interspecific interactions are viewed as a network (*Dáttilo, Díaz-Castelazo & Rico-Gray, 2014*).

In this study, we examine dominance hierarchies for a community of arboreal twig-nesting ants in a coffee agroecosystem. Both arboreal and ground-dwelling twig-nesting ants in coffee agroecosystems are nest-site limited in terms of number (*Philpott & Foster, 2005*), diversity (*Armbrecht, Perfecto & Vandermeer, 2004*; *Gillette et al., 2015*), and size (*Jiménez-Soto & Philpott, 2015*) of nesting resources. For twig-nesting ants, nest takeovers are common, and therefore dominance in this system is defined as competition for nest sites (*Brian, 1952*), and in one case dominance over nest sites has been experimentally demonstrated (*Palmer et al., 2000*).

This present study aims to describe dominance hierarchies for twig-nesting ants due to competition for nest resources in a Mexican coffee agricultural ecosystem. Since

competition is thought to play an important role in the structuring of ant communities, we postulated that resource competition among twig-nesting ants could contribute to the structure of ecological networks involving arboreal ants and their nesting sites. Specifically, we hypothesized that tropical arboreal twig-nesting ants form a linear dominance hierarchy for nesting sites, even when accounting for uncertainties associated with intransitive interactions and sample size. We overall predict that ranking-order remains relatively stable such that higher-ranked individuals maintain their dominant position in the network.

We adopt statistical methods to infer a dominance hierarchy from competitive interactions over nest resources and estimate uncertainty and steepness of that dominance (*Shizuka & McDonald, 2012*; *Pinter-Wollman et al., 2014*; *Sánchez-Tójar, Schroeder & Farine, 2018*). Furthermore, we estimate the orderliness of the hierarchy within the community.

## METHODS

### Study site and system

We conducted fieldwork at Finca Irlanda (15°20′N, 90°20′W), a 300 ha, privately owned shaded coffee farm in the Soconusco region of Chiapas, Mexico with ∼250 shade trees per ha. The farm is located between 900–1,100 m a.s.l (*Perfecto, Vandermeer & Philpott, 2014*). Between 2006–2011, the field site received an average rainfall of 5,726 mm per year with most rain falling during the rainy season between May and October. The farm hosts ∼50 species of shade trees that provide between 30–75% canopy cover to the coffee bushes below. The farm has two distinct management areas—one that is a traditional polyculture and the other that is a mixture of commercial polyculture coffee and shade monoculture coffee according to the classification system of (*Moguel & Toledo, 1999*). Insect collection for this project was authorized under permits from the Secretaria de Medio Ambiente y Recursos Naturales (SEMARNAT) under field study permit numbers 03022, 03696, 03563, 03576, and 05237.

The arboreal twig-nesting ant community in coffee agroecosystems in Mexico is diverse. There are ∼40 species of arboreal twig-nesting ants at the study site including *Brachymyrmex* (3 species), *Camponotus* (8), *Cephalotes* (2), *Crematogaster* (5), *Dolichoderus* (2), *Myrmelachista* (3), *Nesomyrmex* (2), *Procryptocerus* (1), *Pseudomyrmex* (11), and *Technomyrmex* (1) (*Philpott & Foster, 2005*; *Livingston & Philpott, 2010*).

### 'Real-estate' experiments

We examined the relative competitive ability of twig-nesting ants by constructing dominance hierarchies based on 'real estate' experiments conducted in the lab. We collected ants during systematic field surveys in 2007, 2009, 2011, and 2012 in the two different areas of the farm, and then used ants in lab experiments.

Once in the lab, we selected two twigs, each hosting a different species, removed all ants (i.e., all workers, alates and brood) from the twigs and placed them into sealed plastic tubs with one empty artificial nest (15 cm high by 11 cm diameter cylindrical tubs). The artificial nest, or 'real estate', consisted of a bamboo twig, 120 mm long with a 2–4 mm opening. All trials started between 12–2 pm and after 24 h, we opened the bamboo twigs to

note which species had colonized the twig. All ants collected and brought to the lab were used in 'real estate' trials within two days of collection, or were otherwise discarded.

We conducted trials between pairs of the ten most common ant species encountered during surveys: *Camponotus abditus, Camponotus (Colobopsis)* sp. 1, *Myrmelachista mexicana, Nesomyrmex echinatinodis, Procryptocerus scabriusculus, Pseudomyrmex ejectus, Pseudomyrmex elongatus, Pseudomyrmex filiformis, Pseudomyrmex* PSW-53, and *Pseudomyrmex simplex.* We selected a priori to use the 10 most common species and did not run trials between other ant species. We replicated trials for each species pair on average 5.73 times (range: 1–18 trials per pairs of species); four species pairs were replicated once, nine species pairs were replicated twice, and 31 species pairs were replicated three or more times. Only one species pair (*M. mexicana* and *P. filiformis*) was not tested. We conducted 42 trials in 2007, 105 trials in 2009, 82 trials in 2011, and 30 trials in 2012 for a total of 259 trials (Supplementary Materials).

## Dominance hierarchy

We used the trial outcomes to infer the dominance hierarchy and estimate the level of uncertainty and steepness. All simulations were conducted in R version 3.3.3 (*R Core Development Team, 2017*). We used the R package "aniDom" version 0.1.3 to infer dominance hierarchies using the randomized Elo-rating method (*Farine & Sánchez-Tójar, 2017*; *Sánchez-Tójar, Schroeder & Farine, 2018*). To analyze competitive interactions we used the R package "compete" version 0.1 and graphics were completed in the "igraphs" package version 1.2.4.1 (*Csardi & Nepusz, 2006*; *Curley, 2016*).

We subsampled the observed data to determine whether the population had been adequately sampled to infer reliable dominance hierarchies. The subsampling procedure consists of estimating the randomized Elo-rating repeatability values as more data is added to determine if the repeatability values remain stable or decline. Thus, the repeatability values provide insights into the steepness of the hierarchy (*Sánchez-Tójar, Schroeder & Farine, 2018*).

Additionally, we also calculated the ratio of interactions to species to determine sampling effort. An average sampling effort ranging from 10–20 interactions is sufficient to infer hierarchies in empirical networks (*Sánchez-Tójar, Schroeder & Farine, 2018*). We estimated the dominance hierarchy using the randomized Elo-rating method. The matrix of interactions was converted to a sequence of interactions 1,000 times such that different species individual Elo-ratings were calculated each time to obtain mean rankings. We estimated uncertainty in the hierarchy by splitting our dataset into two halves and estimated whether the hierarchy in one half of the matrix correlated with the hierarchy of the other half of the matrix (*Sánchez-Tójar, Schroeder & Farine, 2018*).

In addition to examining the role of ant species attributes and levels of uncertainty in dominance hierarchies, we examined the formation of dominance hierarchies using motif analysis to identify network structures composed of transitive and cyclical triads (*Faust, 2007*). Motif analysis is commonly used in social network analysis to detect emergent properties of the network structure by comparing the relative frequencies of motifs in the observed network to the expected value for the null hypothesis of a random network

(*Holland & Leinhardt, 1972*; *Faust, 2007*). We carried out motif analysis with customized randomization procedures (*McDonald & Shizuka, 2013*) to compare the structure of our network model against random network graphs. Species interaction data were represented as a network plot of the dominance interactions between the 10 species (Fig. 1). The nodes in the network represent ant species and the one-way directional arrows of the edges represent dominant-subordinate relationships. In the random networks, we maintained the same number of nodes and edges as in the observed network, but the directionality and placement of edges were generated randomly. Using the adjacency matrix, we calculated the triad census (*Shizuka & McDonald, 2012*; *McDonald & Shizuka, 2013*). The triad census allows us to examine directed species interactions (*Pinter-Wollman et al., 2014*). We used the seven possible triad configurations fully composed of three nodes that either have asymmetric or mutual edges (*Holland & Leinhardt, 1972*). The triad census can be used to formalize competitive networks into transitive triads (e.g., species A dominates both species B and C) versus cyclic triads (e.g., species A dominates species B, species B dominates species C, and species C dominates species A). The triads are then compared to the null model of random networks.

## RESULTS

### 'Real estate' experiments

Across the vast majority of the trials, there was a clear winner of the 'real estate' battle after 24 h, meaning that one of the two species had occupied the artificial nest. From examining the wins and losses, a clear hierarchy emerged, with some species winning the vast majority of trials in which they were involved, and other species winning few trials. The ranking shows that the twig-nesting species *Myrmelachista mexicana* is the highest ranked species, while *Pseudomyrmex ejectus* is the lowest ranked species in the hierarchy (Table 1). The one trial that did not result in a winner was a trial involving *P. elongatus* and *P. ejectus*.

### Dyadic interactions: Estimating Dominance Hierarchy Uncertainty

The total number of interactions among the 10 species was 258. The ratio of interactions to species (25.8) shows an adequate sampling effort beyond the 10–20 recommended range (*Sánchez-Tójar, Schroeder & Farine, 2018*). Using the randomized Elo-rating method, we found that the hierarchy was intermediate in steepness showing that rank in the hierarchy largely predicts the probability of winning an interaction (Fig. 2). The Elo-rating repeatability was 0.578 which also indicates an intermediate level of uncertainty. We further estimated the uncertainty in the hierarchy by splitting the database into two, and estimating whether hierarchy from one half resembles the hierarchy estimated from the other half. We find that the degree of uncertainty/steepness in the hierarchy is intermediate (mean = 0.43, 2.5% and 97.5% quantile = (−0.12, 0.85)).

### Triad census analysis

The triad census analysis of the triad distribution showed that the observed network has a significant excess of transitive triads followed by a significant deficit of cyclical triads ($T_{tri}$ = 0.66, *p*-value = 0.002). Triad types that are positive (i.e., non-overlapping at 0) occurred
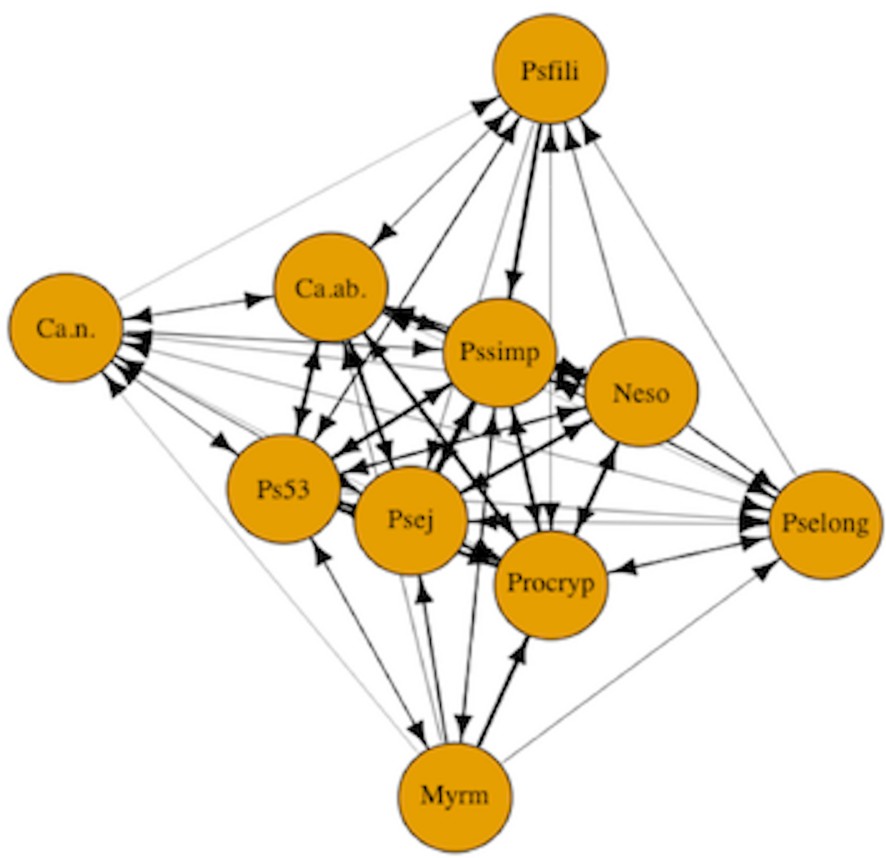

**Figure 1  Competitive network of arboreal ants.** The nodes in the network represent all 10 arboreal ant species and the one-way directional arrows of the edges represent dominant-subordinate relationships. Species are as follows: Myrm = *Myrmelachista mexicana*, Ps53 = *Pseudomyrmex PSW-53*, Neso = *Nesomyrmex echinatinodis*, Ca.ab. = *Camponotus abditus*, Ca.n. = *Camponotus (Colobopsis)* sp. 1, Psfili = *Pseudomyrmex filiformis*, Pssimp = *Pseudomyrmex simplex*, Procryp = *Procryptocerus scabriusculus*, Pselong = *Pseudomyrmex elongatus*, Psej = *Pseudomyrmex ejectus*.

in excess in the observed network, while triad types that are negative showed a deficit in the observed network as compared to the random null network (Fig. 3).

The remaining five triads in the network did not show any significant differences in the mean triad percentage rates between the observed and expected network. While the data showed a clear excess of transitive triangles (34.55%) and deficit for cyclical triangles (3.6%), the distribution for pass-along triads shows a less typical pattern with the 95% confidence intervals crossing the zero line but the mean percentage still showing a deficit.

## DISCUSSION

In this study, we used a novel set of statistical approaches to determine that tropical twig-nesting ants competing for nesting resources are arranged in a linear dominance hierarchy. Although many studies have documented ant dominance hierarchies, it is important to note that ranking methods vary considerably among studies (*Stuble et al.,*
**Table 1** **Estimation of dominance hierarchy using Elo-rating method.** The ranking shows that the twig-nesting species *Myrmelachista mexicana* is the highest ranked species, while *Pseudomyrmex ejectus* is the lowest ranked species in the hierarchy.

| Species | Rankings |
| --- | --- |
| *Myrmecalista mexicana* | 1.402 |
| *Pseudomyrmex (PSW-53)* | 3.833 |
| *Nesomyrmex echinatinodis* | 3.859 |
| *Camponotus abditus* | 5.008 |
| *Camponotus (Colobopsis)* species 1 | 5.173 |
| *Pseudomyrmex filiformis* | 5.517 |
| *Procryptocerus scabriusculus* | 6.911 |
| *Pseudomyrmex simplex* | 7.091 |
| *Pseudomyrmex elongatus* | 7.930 |
| *Pseudomyrmex ejectus* | 8.303 |

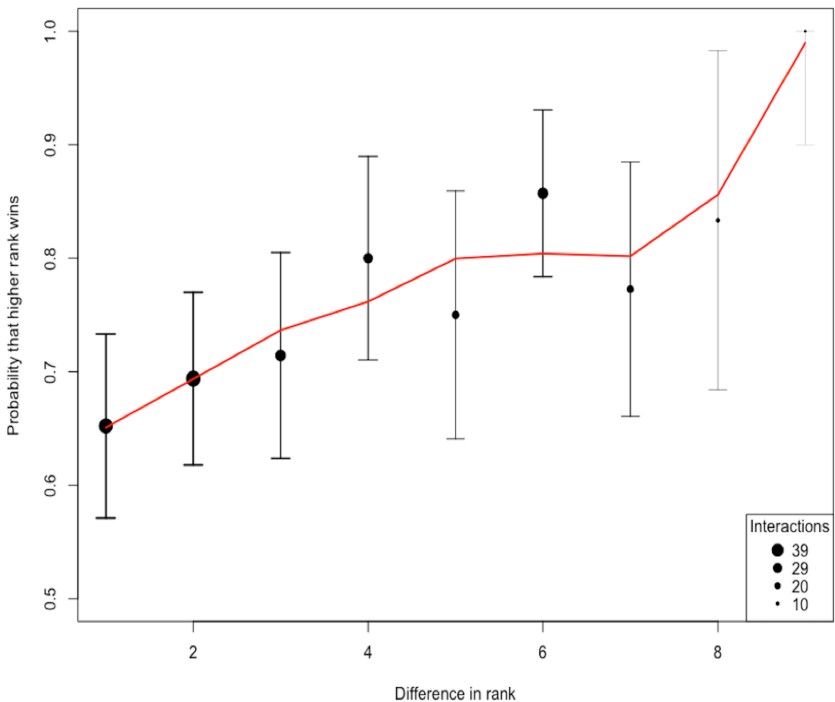

**Figure 2** **The probability of a higher ranked species winning.** The shape of the hierarchy indicates that the rank is intermediate. We quantified the uncertainty/steepness of the hierarchy based on Elo-rating repeatability which is independent of group size and the ratio of interactions to species (*Sánchez-Tójar, Schroeder & Farine, 2018*). Based on the Elo-rating, we find that the value obtained is 0.578 which corroborates our qualitative results showing that the hierarchy is intermediate. Thus, rank in this network is a relatively good predictor that a higher ranked species is more like to win from lower-ranked species even though that is not always the case.

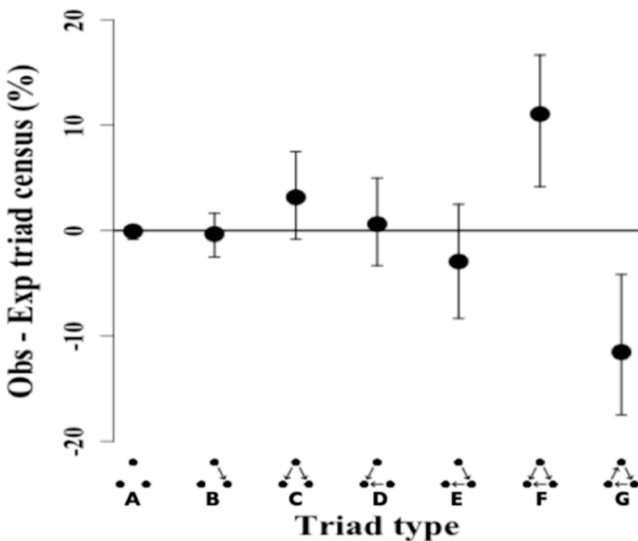

**Figure 3 Triad census of twig-nesting arboreal ants.** We determined the orderliness of hierarchy by estimating the transitivity of interactions. The *y*-axis represents the mean difference between the observed (ten ant species network) and expected (10,000 random networks) percentage of the triad subtypes (shown on the *x*-axis) and error bars show 95% confidence intervals. The twig-nesting ant data shows a significant excess of transitive triads (Tri = 0.66, *p*-value = 0.002) and a significant deficit of cyclical triads. All the other triad sub-types found were not significantly different from the expected random network (zero horizontal line). The following symbols define seven possible triad types: A = Null, B = Single-edge, C = Double-dominant, D = Double-subordinate, E = Pass-along, F = Transitive, G = Cycle. The classic transitive triads are represented by the Double-dominant, Double-subordinate, and Transitive triangles. The Pass-along triad can either turn transitive or cyclical if the third edge becomes established.

*2013*). Traditionally, field studies have quantified dominance relationships on the basis of proportion of contests won. Other studies have use more sophisticated methods to account for competitive reversals (*Vries, 1998*) or have updated rankings based on relative wins and losses during contests (*Colley, 2002*). In this study, we used the randomized Elo-rating by calculating the mean of species Elo-ratings (*Sánchez-Tójar, Schroeder & Farine, 2018*). With this method, we find that the probability of a higher ranked species winning a contest against a lower ranked species is relatively high, which corroborates our finding that the hierarchy has intermediate steepness.

Moving beyond simple pair-wise interactions, we used motif analysis of the network to infer a significant excess of transitive interactions. Transitive interactions were significantly over-represented in the network. Thus the combination of techniques allowed us to determine that the dominance hierarchy in this community is intermediate in steepness and transitive.

Dominance hierarchies over food resources have been commonly documented in ant communities in a variety of ecosystems, but may vary depending on environmental conditions or the amount of food resource provided. For instance, in Mediterranean ecosystems, dominant and subordinate ants are partitioned on the basis of their life-history traits (*Arnan, Cerdá & Retana, 2012*). Dominant ant species had more abundant colonies and displayed increased defense for resources in contrast to subordinate ant

species. Meanwhile, subordinate ants exemplified greater tolerance to higher temperatures (*Cros, Cerdá & Retana, 1997*; *Cerdá, Retana & Manzaneda, 1998*). In addition, outcomes of interspecific interactions within the dominance hierarchy are contingent on environmental conditions (*Arnan, Cerdá & Retana, 2012*). In a temperate forest ecosystem of North Carolina, dominance was context dependent (*Stuble et al., 2017*). Rankings on the basis of food bait monopolization revealed that dominance correlated positively with relative abundance since the most abundant species were ranked higher in the dominance hierarchy. In contrast, rankings based on aggressive encounters did not correlate with abundance. In some habitats, dominance patterns are largely determined by the time of day that foraging occurs (*Bestelmeyer, 2000*). In the North Carolina system, the most abundant ant species, *Aphaenogaster rudis*, was most active during the morning hours, whereas the cold-tolerant ant species, *Prenolepis imparis*, was dominant during the night hours (*Stuble et al., 2017*). Species rankings can also strongly depend on the size of food resources provided in trials. In an assemblage of woodland ants, smaller-sized ants were more efficient at acquiring and transporting fixed resources and larger-sized solitary ants excelled at retrieving smaller food that were mobile during competitive interactions (*LeBrun, 2005*). However, the introduction of phorid parasitoids in this system reduced the transitive hierarchy facilitating the coexistence of subdominant ants (*LeBrun, 2005*; *Lebrun & Feener, 2007*). In our study on competition for nesting sites in the lab, we were able to use fixed resources and to a certain degree control variation in colony size.

It has been suggested that ant dominance hierarchies may be limited in their ability to provide insights into community structure and species coexistence because ranking methods might not be directly related to resource acquisition (*Gordon, 2011*; *Cerdá, Arnan & Retana, 2013*). Furthermore, dominance ranking methods can lead to variation in hierarchies due to inadequate sample sizes (*Stuble et al., 2017*). To account for steepness/uncertainty associated with our ranking methods, we estimated the sampling effort by determining the ratio of species interactions to individuals (*Sánchez-Tójar, Schroeder & Farine, 2018*). The average sampling effort that we found falls within the recommended range reported in the literature (*McDonald & Shizuka, 2013*). The steepness/uncertainty in the hierarchy, independent of both group size and ratio of interactions to individuals, indicates that our ranking approach is robust and representative of the underlying community structure. However, species coexistence can be maintained because subordinate individuals can occasionally outcompete higher-ranked individuals over resources.

Although the twig-nesting ant community that we studied here in lab experiments showed a strong dominance hierarchy, there were some factors that could not be explicitly considered. Species with larger colony sizes might have a competitive advantage over other species. For instance, large colony sizes of invasive Argentine ants are indicative of strong competitive abilities relative to native species (*Holway, 1999*). However, smaller ant colonies can sometimes overtake larger colonies depending on competitive traits, such as chemical defenses in the example of African Acacia ants (*Palmer, 2004*). Although there might be some colony to colony variation in the number of individuals used in each trial (unpublished data), the focus of our study did not involve ant colony size

variation. Preferences for nest entrance sizes is another important consideration that can determine competitive outcomes (*Powell et al., 2011*; *Jiménez-Soto & Philpott, 2015*). While it is certainly the case that ant species prefer different nest entrance sizes, the distribution of natural nest sizes for most of our species (7 out of 10) are statistically indistinguishable. One notable exception is the arboreal ant *P. scabriusculus* which tends to prefer slightly larger nest entrances (in the field) than we provided in the real estate experiments (*Livingston & Philpott, 2010*). However, we have found that *P. scabriusculus* nests in twigs as small as 2–3 mm in diameter.

Dominance hierarchies are often highly context-dependent and species ranking may vary across geographical regions or disturbance regimes (*Palmer, 2004*). Previous research involving ant competition for variable resources in temperate ecosystems showed that intransitive competitive interactions at local spatial scales mediates ant coexistence (*Sanders & Gordon, 2003*). Microclimatic factors also disrupt dominance hierarchies. For instance, environmental variation in coffee systems is likely to influence dominance hierarchies (*Philpott & Foster, 2005*; *Perfecto & Vandermeer, 2011*; *Castillo-Guevara et al., 2019*). Occurrence of fire can disrupt dominance hierarchies in specialist ants in *Acacia* trees resulting in increased abundance of subordinate ants (*Sensenig et al., 2017*). Top down processes such as predation and parasitism likely mediate twig-nesting ant competition in natural communities (*Philpott et al., 2004*; *Feener et al., 2008*; *Hsieh & Perfecto, 2012*). In addition, competition and disturbance from ground- and arboreal carton-nesting ants may influence the colonization and community composition of arboreal twig-nesting ants (*Philpott, 2010*; *Ennis & Philpott, 2017*). Therefore, more comparative research is needed to examine how variable field conditions may affect the hierarchy and ultimately the distribution and relative abundance of different arboreal, twig-nesting ant species.

In addition to dominance hierarchies, there are other factors that can drive the distribution and co-existence patterns of arboreal ant communities (*Palmer et al., 2000*). For instance, variation in life-history trade-offs can influence dominance patterns. Competition-colonization trade-offs have been identified between competitive colonies expanding into nearby trees and foundress queens establishing new nest sites *Stanton, Palmer & Young (2002)*. Twig-ant communities are strongly influenced by canopy structure and habitat complexity (*Philpott, Serber & De la Mora, 2018*). Tree size correlates positively with ant abundance (*Yusah & Foster, 2016*), species richness (*Klimes et al., 2015*), and composition (*Dejean et al., 2008*). Canopy connectivity, in turn, impacts local species coexistence as lower connectivity decreases species richness and canopy connections augment access to tree resources (*Powell et al., 2011*). Limited access to cavity nesting sites hampers growth and reproduction of arboreal ants (*Philpott & Foster, 2005*) and differences in nest entrance size can (*Philpott & Foster, 2005*) affect abundance and richness of arboreal ant species competing for cavity resources (*Powell et al., 2011*; *Jiménez-Soto & Philpott, 2015*). For some cavity-nesting ants (e.g., species in the genus *Cephalotes*), nest entrance size impacted survival and colony fitness (*Powell, 2009*) with important implications for changes in relative abundance over time. Therefore, translating lab competitive hierarchies for nesting sites to ant species co-existence and abundance patterns is not straightforward, but needs to be viewed while considering other factors that simultaneously drive patterns of

distributions and diversity. Subsequent studies should link dominance patterns with relative abundance patterns in the field in order to assess if particular species traits are important in structuring local communities. While competitive outcomes in our experiment are static, dominance hierarchies exhibit considerable variation and field studies should therefore include spatial and temporal variation. Dominance hierarchy studies are typically designed to assess antagonistic interactions, but less focus has been placed on collecting data with neutral interactions (*Stuble et al., 2017*). Differences in food preference and temporal foraging patterns suggest that neither species alter their behavior in the presence of the other. Therefore, more studies noting neutral interactions will shed greater light on the prevalence of dominance hierarchies under natural conditions.

## CONCLUSION

Interspecific dominance hierarchies have been used to explain species coexistence and community structure. One major challenge in the study of ant communities is that different ranking methods have been used to construct dominance hierarchies. In particular, uncertainties associated with species interactions and sampling size are often not quantified. The present study corroborates the existence of dominance hierarchies among tropical arboreal twig-nesting ants. Our study quantified the uncertainty associated with competitive interactions for nesting sites. We show that the shape of the hierarchy is intermediate in steepness, with *Myrmelachista mexicana* ranked highest in the ranking, while *Pseudomyrmex ejectus* was ranked as the lowest in the hierarchy. While lower-ranked individuals can sometimes overtake nesting sites from higher-ranked individuals, the ranking order remains relatively stable. Our analysis of the competition network finds that the hierarchy at the community level is overwhelmingly transitive, suggesting that intransitive interactions are less important in this system. This study contributes to our understanding of the role of competition on the structure of ant communities and dominance hierarchies.

## ACKNOWLEDGEMENTS

The following people assisted with field and lab data collection: G. Domínguez Martínez, U. Pérez Vasquez, G. López Bautista, F. Sanchez-López, D. López, P. Bichier, B. Chilel, A. De la Mora, D. Gonthier, G. Livingston, K. Mathis, K. Ennis, E. Jiménez-Soto, J. Vandermeer, I. Perfecto, D. Jackson, H. Hsieh, and A. Iverson. J. Rojas and E. Chamé Vasquez of El Colegio de la Frontera Sur (ECOSUR) provided logistical support. Permission for arthropod collection was granted by SEMARNAT (Secretaria de Medio Ambiente y Recursos Naturales). We thank Finca Irlanda and Don Walter Peters for access to the farm and housing for field research. We also wish to thank the participants of the NIMBios workshop on ''Animal Social Networks''.

### Funding

This work was supported by the National Science Foundation (No. 1262086). The funders had no role in study design, data collection and analysis, decision to publish, or preparation of the manuscript.

### Grant Disclosures

The following grant information was disclosed by the authors:
National Science Foundation: 1262086.

### Competing Interests

The authors declare there are no competing interests.

### Author Contributions

- Senay Yitbarek performed the experiments, analyzed the data, contributed reagents/materials/analysis tools, prepared figures and/or tables, authored or reviewed drafts of the paper, approved the final draft.
- Stacy M. Philpott conceived and designed the experiments, performed the experiments, authored or reviewed drafts of the paper, approved the final draft.

### Field Study Permissions

The following information was supplied relating to field study approvals (i.e., approving body and any reference numbers):

Insect collection for this project was authorized under permits from the Secretaria de Medio Ambiente y Recursos Naturales (SEMARNAT). The field study permit numbers are 03022, 03696, 03563, 03576, and 05237.

### Data Availability

Data is available at Dryad
https://doi.org/10.5061/dryad.1t0s20m.

### Supplemental Information

Supplemental information for this article can be found online at http://dx.doi.org/10.7717/peerj.8124#supplemental-information.

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
