# Peer review of "Arboreal twig-nesting ants form dominance hierarchies over nesting resources"

_PeerJ, doi:10.7717/peerj.8124_

## Round 0.1 · original submission · Minor Revisions

The reviewers both agree that the manuscript is well-written and scientifically sound, but have both provided a number of suggestions to improve its clarity. Both of them suggested that the motivation for the study and the underlying hypotheses could be more clearly stated. I think that after these revisions the manuscript may be suitable for publication.

·

Basic reporting

This manuscript presented by Yitbarek and Philpott presents a very comprehensive and novel statistical analysis of the dominance hierarchies of arboreal twig-nesting ants over nesting resources. The most prominent shortcoming that I find in this manuscript is the lack of focus towards emphasizing the significance of the findings of the study and their contribution for filling the identified research gap. The effort displayed by the authors in working out a comprehensive and novel statistical method for quantitatively analysing the dominance hierarchies of arboreal twig-nesting ants is highly commendable. However, I observe that the authors have not pitched the manuscript in a way that emphasizes the significance of these findings and their contribution for filling the identified research gap. Provided that these minor revisions will be addressed upon submission, I recommend this manuscript for publication.

As an immediate revision, I suggest the title to be changed as “Arboreal twig-nesting ants form dominance hierarchies over nesting resource”, since this study has focused only on twig-nesting ant species. The existing title, in my opinion, covers a wider range which includes all sorts of arboreal ants, not just twig-nesting ants.

The results that the authors have obtained for dominance hierarchy uncertainty/steepness are quite an interesting and significant finding. To enhance the weight of this finding I suggest including a more detailed description of steepness and how it contributes to community structure and species coexistence in the discussion section. For example, Sánchez-Tójar et al., 2017 provides a descriptive insight of steepness as a source of uncertainty of dominance hierarchy (Sánchez‐Tójar, A., Schroeder, J. and Farine, D.R., 2018. A practical guide for inferring reliable dominance hierarchies and estimating their uncertainty. journal of animal ecology, 87(3), pp.594-608).

Experimental design

While appreciating the effort of authors for identifying the significant research questions existing, I suggest supporting the existence of these research gaps with appropriate citations. For example, Stuble et al., 2017 provides a good insight of the existing knowledge and limitations involved with this research question (Stuble, K.L., Jurić, I., Cerda, X. and Sanders, N.J., 2017. Dominance hierarchies are a dominant paradigm in ant ecology (Hymenoptera: Formicidae), but should they be? And what is a dominance hierarchy anyways. Myrmecological News, 24, pp.71-81.).

I agree with the authors’ point presented in lines 285-290, that there can be factors other than dominance hierarchy, influencing distribution and coexistence of arboreal ant communities. I suggest that this statement will carry more weight in the manuscript if a suitable regression analysis can be conducted to determine to what percentage does ant distribution and coexistence depend on dominance hierarchies. According to the information given in 116-118 I presume that the data collected from surveys on ant abundance/distribution and the dominance hierarchy rankings stated in Table 1 will serve this purpose.

Validity of the findings

Since the beginning of the manuscript the authors’ main objective had been to present novel statistical analyses to explain dominance hierarchies of arboreal twig-nesting ants over nesting species. Hence, the paragraph coming under discussion section (lines 228-251), elaborating existing knowledge on dominance hierarchies over food resources in ant communities fails to make a direct connection with the results obtained by the authors and the ultimate research question addressed by the manuscript. I suggest further elaboration on resource utilization in nesting sites and connection of that information with the results obtained on competition shown by different species over nesting sites. For example, Adams et al., 2019 provides some insight into how resource availability in nesting sites contribute to community structure. (Adams, B.J., Schnitzer, S.A. and Yanoviak, S.P., 2019. Connectivity explains local ant community structure in a Neotropical forest canopy: a large‐scale experimental approach. Ecology, 100(6), p.e02673.) In fact, the information included in lines 290-303, on how nest site factors such as canopy structure, tree size and nest entrance size, gives a comprehensive explanation on how dominance hierarchies over nesting sites affect the species coexistence, based on existing knowledge and findings as well as existing limitations and knowledge gaps. Hence, I suggest combining these two sections. It will better establish the relationship between dominance hierarchy and variability observed in the results in resource utilization in nesting sites, shown by coexisting species arboreal twig-nesting ant species. This relationship will add more weight to the answers provided by this study when addressing the major research question of how dominance hierarchies contribute to species coexistence.

The authors have presented a very interesting and novel approach for quantifying dominance hierarchies in arboreal ant communities and have made a good attempt to explain species coexistence in ant communities through it. However, in the conclusion section, I fail to see the weight and significance of those findings and their contribution to answering the main research question stated at the beginning. Rather, I see background information and suggestions for followup research included in the conclusion, which are content that could be moved to the discussion section. Hence, I suggest re-writing the conclusion in a way that the findings of this study are highlighted and emphasizes its contribution for filling the existing research gap.

Additional comments

I suggest moving the sentence, in lines 168-169, “The nodes in the network represent…” to the figure caption to improve the comprehensibility of the figure.


I commend the authors for their successful attempt of analyzing dominance hierarchies of arboreal twig-nesting ant species using a novel and comprehensive statistical approach, based on extensive field and lab work added to thorough literature review provided. Also I appreciate the manuscript written in unambiguous English. So I recommend this manuscript for publication, given that the above mentioned minor revisions are addressed and the manuscript is pitched in a manner that the significance and contribution of this study is emphasized to make a bigger impact.

Reviewer 2 ·

Basic reporting

In this work, Yitbarek and Phipott analyze whether arboreal twig-nesting ants competing for nesting resources in a Mexican coffee agricultural ecosystem are arranged in a linear dominance hierarchy. The topic is very interesting and timing: there is now an intense debate on the effects of competition and compensatory mechanisms structuring ant communities. Yitbarek and Philpott present an interesting study with timely and novel results. I find the study and results of merit, and are within the scope of PeerJ. However, I see some sections of the manuscript need to be clarified and improvements. The introduction and background to show good context. However, in order to reinforce and enrich the theorical context, it is suggested to attach some references of studies conducted in temperate and tropical environments. The hypothesis is not completely understandable, what is it: to form a "clear hierarchy of dominance" for nesting sites? It is requested to be explicit in the wording of the hypothesis and describe the prediction (s) of the hypothesis. In Methodology I see some terms (triad transitive) need to be clarified. References need to be carefully checked along the entire Ms. General comments:
Page 7, line 55. Delete the dot
Page 8, line 74. It is suggested to add as a background a study conducted in temperate zones with dominance hierarchy: Castillo-Guevara C, Cuautle M, Lara C, Juárez-Juárez B. 2019. Effect of agricultural land-use change on ant dominance hierarchy and food preferences in a temperate oak forest. PeerJ 7: e6255
Page 8, line 76. It is suggested to add as background the studies carried out in tropical areas and plant-ant interaction networks: Díaz-Castelazo C, Rico-Gray V, Oliveira PS, Cuautle M. 2004. Extrafloral nectary-mediated ant-plant interactions in the coastal vegetation of Veracruz, Mexico: richness, occurrence, seasonality and ant foraging patterns. Ecoscience 11: 472-481.
Page 8, line 80. It is suggested to add as a background a study carried out in tropical areas, dominance hierarchy and plant-ant interaction networks: Dáttilo W, Díaz-Castelazo C, Rico-Gray V. 2014. Ant dominance hierarchy determines the nested pattern in ant-plant networks. Biological Journal of the Linnean Society 113 (2): 405-414.
Page 9, line 94. It is suggested to be explicit in the wording of the hypothesis and to describe the prediction (s) of the hypothesis.

Experimental design

The method with which the question is addressed is considered very original, applies this approach to an ecological study in an authentic and unique way. However, I see some terms (triad configurations, explicitly explain triad transitive) need to be clarified. General comments:
Pag. 9, line 109. It is suggested to mention at least the taxonomic category of subfamily to which belong the 10 species selected for the study.
Pag. 10, line 129. Pseudomyrmex PSW-53, explain this key or name.
Page 12, line 173. Develop the seven possible triad configurations. Figure 3 refers to the triad sub-types (A-G) only with symbols, but the definition of each type is not clear.

Validity of the findings

The information provided by this research is considered of good quality, the contribution in the field of ecology of the ant community is fortunate, since the study of the dominance hierarchy has been controversial and the information generated from the novel experiments in the laboratory it is an advance in this field. In addition, where it is carried out, the coffee agroforestry system is an ecosystem rich in species and interactions that has been little studied at this level of networks. General comments:
Page 14, line 225. Explain the biological consequences of a transitive triad, that is, its biological significance, giving an example with ant species, how they are expected to behave in relation to nest competition.
Page 15, line 272. To reinforce this part of other factors that influence the dominance hierarchy of ants, it is suggested to mention another study (Castillo-Guevara et al. 2019) that includes experiments in natural conditions in temperate zones.

Additional comments

General comments.
The specific corrections of the Ms are indicated in the PDF.
References. References need careful editing, some quotes are not in the references and vice versa. Some references have erroneous information, it is suggested to check it carefully. The PeerJ format of the references carries the DOI. Add the DOI to all references. The specific corrections of the references are indicated in the PDF.
Figure 3. In the text explain the triad sub-types (A-G) that are represented with symbols in the figure.

Annotated reviews are not available for download in order to protect the identity of reviewers who chose to remain anonymous.

---

## Round 0.2 · accepted · Accept

In the previous round of review the Reviewers made a number of relatively minor suggestions to improve the flow and argumentation of the argument. I believe that the authors have addressed all of these issues, and the final manuscript is both easier to read and stronger overall.